# Clinical and Virological Features of SARS-CoV-2 Variants during the Four Waves of the Pandemic in the Mexican Southeast

**DOI:** 10.3390/tropicalmed8030134

**Published:** 2023-02-23

**Authors:** Guadalupe del Carmen Baeza-Flores, Juan Pedro Luna-Arias, Jesús Arturo Ruiz-Quiñones, Xavier Miguel Boldo-León, Alberto Cedro-Tanda, Dora Garnica-López, Alfredo Mendoza-Vargas, Jesús M. Magaña-Cerino, Mirian Carolina Martínez-López

**Affiliations:** 1Molecular Diagnostic Laboratory, Research Center, Academic División of Health Sciences (DACS), Juarez Autonomous University of Tabasco (UJAT), Villahermosa 86150, Mexico; 2Center for Research and Advances Studies, Department of Cell Biology, The National Polytechnic Institute (CINVESTAV-IPN), Mexico City 07360, Mexico; 3Research Center for Tropical and Emerging Diseases, High Specialty Regional Hospital “Dr. Juan Graham Casasus”, Villahermosa 86126, Mexico; 4National Institute of Genomic Medicine, Mexico City 14610, Mexico

**Keywords:** SARS-CoV-2, variants, pandemic waves

## Abstract

We conducted a retrospective study using a population of patients who were hospitalized at Dr. Juan Graham Casasus Hospital in Villahermosa (Tabasco, Mexico) and had a positive RT-PCR test for SARS-CoV-2 between June 2020 and January 2022. We analyzed all medical records, including demographic data, SARS-CoV-2 exposure history, underlying comorbidities, symptoms, signs at admission, laboratory findings during the hospital stay, outcome, and whole-genome sequencing data. Finally, the data were analyzed in different sub-groups according to distribution during waves of the COVID-19 pandemic regarding Mexican reports from June 2020 to January 2022. Of the 200 patients who tested positive via PCR for SARS-CoV-2, only 197 had samples that could be sequenced. Of the samples, 58.9% (n = 116) were males and 41.1% (n = 81) females, with a median age of 61.7 ± 17.0 years. Comparisons between the waves of the pandemic revealed there were significant differences in the fourth wave: the age of patients was higher (*p* = 0.002); comorbidities such as obesity were lower (*p* = 0.000), while CKD was higher (*p* = 0.011); and hospital stays were shorter (*p* = 0.003). The SARS-CoV-2 sequences revealed the presence of 11 clades in the study population. Overall, we found that adult patients admitted to a third-level Mexican hospital had a wide range of clinical presentations. The current study provides evidence for the simultaneous circulation of SARS-CoV-2 variants during the four pandemic waves.

## 1. Introduction

The pandemic caused by severe acute respiratory syndrome coronavirus-2 (SARS-CoV-2) rapidly spread from Wuhan (Hubei, China) to other countries [1] and was first detected in Mexico in February 2020. The periods of time that occurred from the start of the pandemic have been described as waves. Concerning these periods in Mexico, the first wave spanned epidemiological weeks 8 to 39 of 2020; the second wave lasted from epidemiological week 40 of 2020 to week 15 of 2021; the third wave occurred in weeks 23 to 42 of 2021; and finally, the fourth wave occurred between epidemiological week 51 of 2021 and week 9 of 2022 [2].

The most typical clinical manifestations of COVID-19 include cough, fever, dyspnea, and pneumonia. The WHO (World Health Organization) has classified cases into mild, moderate, and severe. The diagnosis of COVID-19 is confirmed by the current gold standard SARS-CoV-2 diagnostic tool, which is real-time reverse-transcription polymerase chain reaction (RT-PCR) [3] and/or enzyme immunoassay testing [4,5].

Two years into the pandemic, numerous variants of the SARS-CoV-2 have emerged across the world. These mutants have circulated across many countries and have been held responsible for abrupt infectious waves. They have also been categorized as variants of interest (VOI) and/or variants of concern (VOC). The emergence of novel VOC requires rapid genomic, virological, epidemiological, and clinical characterization to inform public health, clinical, and research responses [6]. Adaptive mutations in the viral genome can alter the pathogenic potential of the virus. In light of this situation, enormous efforts have been carried out to understand the evolution and genomic diversity of SARS-CoV-2, as well as the risk of disease severity in terms of the risk of hospitalization, ICU admission, and mortality [7].

Interestingly, it has been suggested that the phenotypic difference in hospitalized patients are related to the variants associated with different stages (or waves) of the COVID-19 pandemic [8]. A difference in behavior between pandemic waves has recently been reported, where the third wave reported in Spain involved less use of mechanical ventilation and, as a consequence, lower incidence of nosocomial infections, complications, length of stay in the ICU, and mortality [9]. In the last few months, there have been several studies focused on the severity of the disease, differences in symptoms, and the attitudes of the people through the first three waves of the COVID-19 pandemic with regard to new variants produced by genomic mutations [10].

In this study, we aimed to determine the demographic features, disease severity, and clinical outcomes during the different pandemic waves in Mexican patients. Additionally, we investigated the circulating variants from 2020 until January 2022 in patients admitted to a third-level southeast Mexican hospital. 

## 2. Materials and Methods

### 2.1. Study Population

This retrospective study included patients hospitalized at Dr. Juan Graham Casasus Hospital in Villahermosa (Tabasco, Mexico) who tested positively for SARS-CoV-2 between June 2020 until January 2022. This study was registered with the Australian New Zealand Clinical Trials Registry (ACTRN12620001265965) by the ethics committee of Juan Graham Casasus Hospital (CEI/JGC/232021). Informed consent for retrospective data collection was waived.

The study included adult hospitalized patients with one or more symptoms of COVID-19. A positive result was confirmed by the detection of SARS-CoV-2 in a respiratory sample by reverse-transcriptase polymerase chain reaction (RT-PCR) with a cycle threshold (CT) value of less than 25, from June 2020 through January 2022. Patients diagnosed by antigen tests were not included. 

### 2.2. Data Sources 

We analyzed every medical record. Demographic data, SARS-CoV-2 exposure history, underlying comorbidities, symptoms, signs at admission, laboratory findings during hospital stay, and outcome were analyzed in different sub-groups considering the pandemic wave distribution in accordance with Mexican reports (the first wave occurred approximately during epidemiological weeks 8–39 of 2020; the second wave occurred between epidemiological week 40 of 2020 and week 15 of 2021; the third wave occurred during weeks 23–42 of 2021; and the fourth wave occurred between epidemiological week 51 of 2021 and week 9 of 2022) [2].

### 2.3. Samples

The study included a total of 200 nasal and pharyngeal swab samples. The swabs were collected from the patients upon admission to the hospital. Viral transport media were stored at −80 °C. 

Fresh RNA was extracted from the samples. The RNA extraction was developed following the manufacturer’s instructions. In brief, 50 μL of RNA was isolated and purified using an iPrep™ PureLink^®^ Virus Kit (INVITROGEN^®^, Waltham, MA, USA). SARS-CoV-2 was detected by QuantStudio 5 Real-Time PCR (Applied Biosystems, Waltham, MA, USA), and amplification was conducted using a GeneFinder^TM^ COVID19 Real Amp Kit v2 (Cat IFMR-45; OSANG Healthcare Co., Ltd., Anyang, Republic of Korea). The GeneFinder^TM^ kit was designed to detect RdRp, N, and E genes from SARS-CoV-2 and an internal control. The settings used for RT-PCR were as follows: 1 cycle at 50 °C for 20 min (reverse transcription); 1 cycle at 95 °C for 5 min (pre-denaturation); and 45 cycles at 95 °C for 15 s (denaturation); and 58 °C for 60 s (annealing). The analysis settings for target RdRp, N, and E genes were threshold = 15,000, baseline, 3–15. Meanwhile, those for IC were threshold = 10,000; baseline, 3–15.

### 2.4. Next-Generation Sequencing (NGS)

All SARS-CoV-2 RNA-positive samples underwent real-time whole-genome sequencing at the National Institute of Genomic Medicine in Mexico City, using an Illumina MiSeq platform.

### 2.5. Statistical Analysis

Sociodemographic variables were collected, and we performed group analyses using the Chi-squared test or Fisher’s exact test for categorical variables, with the results expressed as frequencies. For continuous variables, we conducted one-way analysis of variance (ANOVA), with the significance level set at *p* < 0.05. The results are presented as the mean ± error of the mean. We analyzed these data using SPSS v21 software (IBM^®^ SPSS^®^ Statistics).

## 3. Results

From June 2020 to January 2022, of the 200 patients who tested positive via RT-PCR for SARS-CoV-2, only 197 had samples that could be sequenced. Of the samples, 58.9% (n = 116) were males and 41.1% (n = 81) were females, with a median age of 61.7 ± 17.0 years. We determined that the percentage of patients with COVID-19 per wave was as follows: 18.3% (n = 36) in the first wave; 35% (n = 69) in the second wave; 18.3% (n = 36) in the third wave; and 28.4% (n = 56) in the fourth wave. 

Table 1 shows the main clinical features grouped into the four waves. Patients diagnosed in the fourth wave were significantly older than those in the other waves (*p* = 0.002). Gender and number of comorbidities did not differ between the four waves. We found that 73.1% of patients were unvaccinated overall. It should be noted that 48.6% of patients in the first wave had a diagnosis of obesity (*p* = 0.000), while in the fourth wave, 18.4% had chronic kidney disease. The number of symptoms presented differed significantly between the four waves of the COVID-19 pandemic (*p* = 0.001). In particular, during the first wave, the number of patients with fever (*p* = 0.040), cough (*p* = 0.007), odynophagia (*p* = 0.028), and chill (*p* = 0.010) was significantly higher than in the other waves. The number of COVID-19 symptoms concerning vaccination status was analyzed, and there was no differences between groups (*p* = 0.216). Odynophagia (*p* = 0.018) and diarrhea (*p* = 0.005) were more common in unvaccinated patients. The full details are exhibited in Appendix A. 

The outcomes and laboratory parameters are presented in Table 2. The mean duration of hospital stay was 5.6 ± 3.6 days. At admission, the mean count of erythrocytes (*p* = 0.000), hemoglobin (*p* = 0.000), and hematocrit (*p* = 0.000) were lower in the fourth wave than in the other waves. In contrast, the mean of urea (*p* = 0.000), BUN (*p* = 0.000), and creatinine (*p* = 0.013) were higher, while the mean of fibrinogen (*p* = 0.001) was lowest in the second wave. Finally, 56.1% of patients died during their stay. 

Regarding the number of patients who died with a high or low CT, there was no statistically significant difference according to pandemic waves. The full details are exhibited in Appendix A. The analysis of circulating clades is shown in Table 3. The SARS-CoV-2 sequences revealed the presence of 11 clades in the study population. We found that the highest number of samples collected were distributed in the 20B clade (27.4%; n = 54), followed by the 21K (Omicron) clade (26.9%; n = 53), and the 20A clade (19.8%; n = 39). 

From Figure 1, it can be seen that the majority of genomes during the first wave were dominated by the 20A clade, followed by the 20C clade, while the lowest number of cases were attributed to the 20B clade. The second and third waves included 10 different circulating genomes. Finally, the fourth wave mainly consisted of the 21J (Delta) and 21K (Omicron) clades.

## 4. Discussion

Mexico started screening for SARS-CoV-2 in February of 2020. The present study represents the first report comparing the demographic, clinical, and virological features of COVID-19 patients throughout the four pandemic waves in Tabasco, Mexico. 

One limitation of this study is that the inclusion criteria restricted the sample as we only included patients with a positive RT-PCR with a cycle threshold lower than 25. In this sense, we had a similar number of events in the first, second, and third waves, while the last (fourth) wave comprised more cases. There is substantial evidence in the literature supporting the idea that lockdown policies had major implications regarding the pattern of pandemic waves of SARS-CoV-2 [11,12]. For example, a recent study showed differences in behavior due to the number of positive cases progressively decreasing in Israel [13].

We found significant differences in clinical presentations across the four waves. The demographic data indicated that patients in the fourth pandemic wave were older than the others and the proportion of males was higher. At present, to the best of our knowledge, our data are consistent with the previously reported high vulnerability of older men [14]. We found that patients in the third wave were significant younger compared to the fourth wave (53.8 ± 3.5 vs. 67.2 ± 1.2); however, an Indian study observed that patients in the group aged ≥75 years increased in the second wave, and they observed similar behavior in the proportion of females in the second wave [14]. There may also be differing features according to geographic trends; for example, in a South African case study, patients hospitalized during the fourth wave were younger, there was a higher proportion of females, significantly fewer patients with comorbidities were admitted, and the proportion of patients presenting with an acute respiratory condition was lower [11]. 

An Italian study reported that patients in the first wave were more frequently affected by multiple chronic diseases [8]. We reported that in Tabasco, Mexico, patients admitted in the third wave had a higher number of comorbidities. More than 30% of the admitted patients in all waves had hypertension and, in agreement with previous studies, no significant differences between the different waves were observed in Tabasco [15]. Our findings regarding comorbidities showed that, for patients in the first and third waves, the most common comorbidities were obesity and hypertension; in the second wave, diabetes and hypertension were the most common; and in the fourth wave, diabetes, hypertension, and chronic kidney disease (CKD) were the most common. 

Regarding symptoms, patients in the first and third waves had more symptoms than the others. Dyspnea was common to all waves; however, fever, cough, odynophagia, and chill were significantly higher in the first wave than the others. 

With respect to laboratory testing, there were significant differences in the mean count of erythrocytes, hemoglobin, hematocrit, urea (*p* = 0.000), BUN, creatinine, and fibrinogen. A study comparing the first and second waves reported that the mean absolute lymphocyte count was significantly lower, while C-reactive protein and D-Dimer were significantly higher in the second wave [14]. 

The rate of mortality was high in the four waves, with above 55% of patients dying. On the contrary, El Shabasy et al. found that mortality rate was variable between the waves, with a high mortality rate in the first wave and a lower rate in the third wave [10]. In addition, it has been found that overall mortality in Indian patients in the second wave was nearly 40% higher than that in first wave [14]. In agreement with previous reports, the present study confirmed that mutations in the viral genes studied may have had a direct correlation with clinical outcomes according to the pandemic waves [16]. 

In Tabasco, Mexico, during the first wave of the COVID-19 pandemic, three clades were isolated: 20A, 20B, and 20C. In the second wave, there were five circulating clades (20A, 20B, 20C, 20G, 20I Alpha); however, notably, the third pandemic wave presented the highest frequency of circulating clades (20A, 20B, 20I Alpha, 20J Gamma, 21C Epsilon, 21G Lambda, 21I Delta). Finally, the fourth pandemic wave was characterized by two major clades (21J Delta, 21K Omicron). Interestingly, in a study considering 74 samples from African patients, it was observed that the major SARS-CoV-2 variants of concern circulating were Beta in the first and second waves and Delta in the third wave [17].

The difference between our study and that of the study conducted by Lycett et al. was their focus on the spread of dominant SARS-CoV-2 lineages that were introduced into Scotland in the first wave, as well as the association between lockdowns and the reduction in circulating lineages in the second wave [18].

## 5. Conclusions

In this retrospective study, we found that the adult patients admitted to a third-level Mexican hospital with COVID-19 had a wide range of a clinical presentations. The current study provides evidence for the simultaneous circulation of SARS-CoV-2 variants in the four pandemic waves. Finally, we confirmed that the 20B and 21K Omicron clades were the most abundant in the study population.

## Figures and Tables

**Figure 1 tropicalmed-08-00134-f001:**
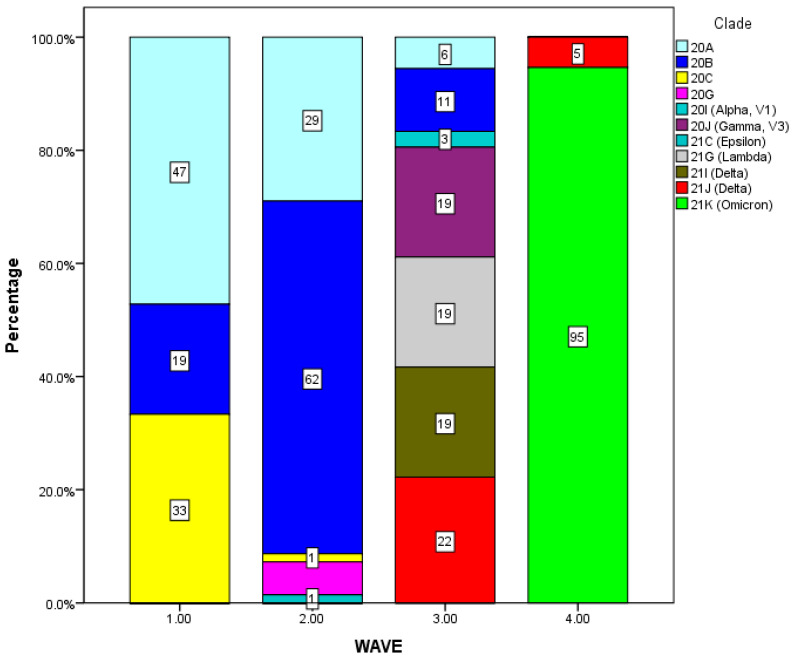
Circulating SARS-CoV-2 clades in the four pandemic waves. The bars illustrate: 1. the first wave (during epidemiological weeks 8–39 of 2020); 2. the second wave (between epidemiological week 40 of 2020 and week 15 of 2021); 3. third wave (during weeks 23–42 of 2021); and 4. the fourth wave (between epidemiological week 51 of 2021 and week 9 of 2022).

**Table 1 tropicalmed-08-00134-t001:** Demographic features of patients with COVID-19 stratified by waves of the pandemic.

	Mean	Wave 1	Wave 2	Wave 3	Wave 4	*p*
Age (years)	61.7 ± 17.0	58.1 ± 2.2	62.6 ± 1.8	53.8 ± 3.5 *	67.2 ± 1.2 *	0.002 *
Male (%, n)	58.9% (n = 116)	55.6% (n = 20)	59.4% (n = 41)	58.3% (n = 21)	60.7% (n = 34)	0.968
Female (%, n)	41.1% (n = 81)	44.4% (n = 16)	40.6% (n = 28)	41.7% (n = 15)	39.3% (n = 22)	
Unvaccinated (%, n)	73.1% (n = 144)	100% (n = 35)	100% (n = 64)	70.0% (n = 21)	49.0% (n = 24)	0.000 *
Incomplete vaccine status (%, n)	2.5% (n = 5)	0% (n = 0)	0% (n = 0)	13.3% (n = 4)	2.0% (n = 1)	
Complete vaccine status (%, n)	14.7% (n = 29)	0% (n = 0)	0% (n = 0)	16.7% (n = 5)	49.0% (n = 24)	
Comorbidities (mean ± EE)	1.3 ± 1	1.49 ± 0.1	1.40 ± 0.1	1 ± 0.2	1.33 ± 0.1	0.275
Obesity (%, n)	24.4% (n = 43)	48.6% (n = 17)	27.0% (n = 17)	24.1% (n = 7)	4.1% (n = 2)	0.000 *
Diabetes (%, n)	32.5% (n = 64)	37.1% (n = 13)	42.9% (n = 27)	17.2% (n = 5)	38.8% (n = 19)	0.118
Hypertension (%, n)	44.3% (n = 78)	48.6% (n = 17)	47.6% (n = 30)	31.0% (n = 9)	44.9% (n = 22)	0.455
HIV (%, n)	1.1% (n = 2)	2.9% (n = 1)	1.6% (n = 1)	0% (n = 0)	0% (n = 0)	0.586
Cardiovascular disease (%, n)	8.0% (n = 14)	2.9% (n = 1)	7.9% (n = 5)	10.3% (n = 3)	10.2% (n = 5)	0.613
CKD (%, n)	8.0% (n = 14)	2.9% (n = 1)	6.3% (n = 4)	0% (n = 0)	18.4% (n = 9)	0.011 *
Symptoms (mean ± EE)	6.4 ± 3.0	7.8 ± 0.5	5.9 ± 0.3	7.2 ± 0.6	5.5 ± 0.3	0.001 *
Fever (%, n)	72.3% (n = 128)	82.9% (n = 29)	76.6% (n = 49)	75.9% (n = 22)	57.1% (n = 28)	0.040 *
Cough (%, n)	85.9% (n = 152)	94.3% (n = 33)	90.6% (n = 58)	89.7% (n = 26)	71.4% (n = 35)	0.007 *
Odynophagia (%, n)	28.2% (n = 50)	48.6% (n = 17)	25% (n = 16)	20.7% (n = 6)	22.4% (n = 11)	0.028 *
Dyspnea (%, n)	75.7% (n = 75.7)	80.0% (n = 28)	73.4% (n = 47)	82.8% (n = 24)	71.4% (n = 35)	0.615
Diarrhea (%, n)	16.4% (n = 29)	22.9% (n = 8)	15.6% (n = 10)	27.6% (n = 8)	6.1% (n = 3)	0.057
Chill (%, n)	7.9% (n = 14)	20.0% (n = 7)	7.8% (n = 5)	2.9% (n = 2)	0 (n = 0)	0.010 *
Headache (%, n)	76.3% (n = 135)	77.1% (n = 27)	76.6% (n = 49)	79.3% (n = 23)	73.5% (n = 36)	0.945
Myalgia (%, n)	46.9% (n = 83)	51.4% (n = 18)	40.6% (n = 26)	41.4% (n = 12)	55.1% (n = 27)	0.395
Joint pain (%, n)	44.6% (n = 79)	45.7% (n = 16)	37.5% (n = 24)	48.3% (n = 14)	51.0% (n = 25)	0.513
Runny nose (%, n)	22.6% (n = 40)	22.9% (n = 8)	23.4% (n = 15)	27.6% (n = 8)	18.4% (n = 9)	0.816
Anosmia (%, n)	10.2% (n = 18)	14.3% (n = 5)	9.4% (n = 6)	17.2% (n = 5)	4.1% (n = 2)	0.234
Ageusia (%, n)	10.2% (n = 18)	14.3% (n = 5)	10.9% (n = 7)	17.2% (n = 5)	2.0% (n = 1)	0.121

CKD: chronic kidney disease; HIV: human immunodeficiency virus infection. * denotes a significant difference at the *p* < 0.05 level.

**Table 2 tropicalmed-08-00134-t002:** Clinical features of patients with COVID-19 stratified by pandemic wave.

	Mean	Wave 1	Wave 2	Wave 3	Wave 4	*p*
Hospital stay (mean days ± EE)	5.6 ± 3.6	9.5 ± 1.4	15.0 ± 1.5 *	12.0 ± 2.0	7.9 ± 0.9 *	0.003 *
Died	56.1% (n = 110)	66.7% (n = 24)	58.0% (n = 40)	48.6% (n = 17)	51.8% (n = 29)	0.398
RdRp gene (CT)	17.9 ± 2.1	18.4 ± 1.5	17.5 ± 1.9	18.0 ± 1.8	18.0 ± 2.9	0.180
N gene (CT)	18.2 ± 2.4	17.8 ± 1.8	17.2 ± 2.2	18.1 ± 2.1	19.1 ± 2.8 *	0.002 *
E gene (CT)	17.2 ± 2.7	15.1 ± 2.2	15.5 ± 2.4	17.3 ± 2.2 *	17.2 ± 2.7 *	0.000 *
Erythrocyte (10^6^/μL)	4.2 ± 0.06	4.4 ± 0.1	4.3 ± 0.08	4.6 ± 0.1	3.7 ± 0.1 *	0.000 *
Hemoglobin (g/dL)	12.4 ± 0.1	13.0 ± 0.3	12.7 ± 0.2	13.9 ± 0.3	10.7 ± 0.4 *	0.000 *
Hematocrit (%)	37.7 ± 0.5	39.3 ± 1.1	38.9 ± 0.7	41.3 ± 0.9	32.6 ± 1.3 *	0.000 *
Leucocytes (10^3^/μL)	12.3 ± 0.5	12.2 ± 0.9	12.4 ± 0.8	11.5 ± 0.8	12.7 ± 1.6	0.930
Lymphocytes (10^3^/μL)	1.7 ± 0.4	0.9 ± 0.09	1.7 ± 0.5	0.9 ± 0.1	2.7 ± 1.5	0.536
Monocytes (10^3^/μL)	0.6 ± 0.0	0.5 ± 0.07	0.6 ± 0.1	0.7 ± 0.1	0.6 ± 0.0	0.938
Eosinophils (10^3^/μL)	0.02 ± 0.0	0.01 ± 0.01	0.03 ± 0.01	0.005 ± 0.0	0.02 ± 0.0	0.374
Basophils (10^3^/μL)	0.01 ± 0.0	0.003 ± 0.003	0.005 ± 0.002	0.02 ± 0.0	0.02 ± 0.0	0.031 *
Neutrophils (10^3^/μL)	10.6 ± 0.6	13.2 ± 2.7	10.8 ± 0.8	10.0 ± 0.8	9.3 ± 0.8	0.229
Thrombocytes (10^3^/μL)	245.4 ± 12.4	227 ± 14.4	268.4 ± 28.6	229.4 ± 17.3	237.6 ± 19.9	0.567
C-reactive protein	189.0 ± 9.3	207.7 ± 24.8	182.8 ± 14.2	172.2 ± 16.8	203.4 ± 22.2	0.559
Glucose (mg/dL)	178.2 ± 7.3	213.6 ± 24.2	183.0 ± 12.3	151.7 ± 12.1	167.8 ± 11.5	0.066
Urea (mg/dL)	60.8 ± 3.7	46.9 ± 4.4	53.6 ± 5.6	42.6 ± 3.3	91.8 ± 9.8 *	0.000 *
BUN (mg/dL)	28.3 ± 1.7	22.8 ± 2.0	25.0 ± 2.6	19.9 ± 1.5	42.1 ± 4.6 *	0.000 *
Creatinine (mg/dL)	2.0 ± 0.4	1.02 ± 0.1	1.40 ± 0.3	1.0 ± 0.1	4.1 ± 1.3 *	0.013 *
Na (mmol/L)	137.3 ± 0.5	136.0 ± 0.6	137.6 ± 1.0	139.5 ± 0.7	136.1 ± 1.0	0.108
K (mmol/L)	4.5 ± 0.05	4.5 ± 0.6	4.5 ± 0.08	4.4 ± 0.1	4.6 ± 0.1	0.583
CL (mmol/L)	100.3 ± 0.5	100.5 ± 1.1	99.6 ± 1.0	102.0 ± 0.8	100.1 ± 1.0	0.504
BILT (mg/dL)	0.7 ± 0.0	0.82 ± 0.1	0.6 ± 0.04	0.6 ± 0.05	1.1 ± 0.3	0.106
BILDir (mg/dL)	0.3 ± 0.0	0.3 ± 0.09	0.2 ± 0.02	0.2 ± 0.0	0.6 ± 0.3	0.133
BILInd (mg/dL)	0.4 ± 0.0	0.4 ± 0.05	0.3 ± 0.02	0.4 ± 0.0	0.4 ± 0.0	0.098
ProtT (g/dL)	7.0 ± 0.3	6.8 ± 0.08	7.5 ± 0.9	6.8 ± 0.1	6.7 ± 0.1	0.845
Albumin (g/dL)	3.5 ± 0.4	3.6 ± 0.8	3.4 ± 0.07	3.6 ± 0.0	3.4 ± 0.0	0.125
Globulin (g/dL)	3.1 ± 0.0	3.2 ± 0.5	3.1 ± 0.05	3.1 ± 0.0	3.2 ± 0.1	0.583
ALT (UI/L)	50.7 ± 5.1	52.1 ± 8.8	41.1 ± 3.0	57.0 ± 9.0	60.1 ± 18.5	0.488
AST (UI/L)	73.5 ± 11.1	56.2 ± 6.4	53.0 ± 3.2	92.0 ± 26.4	104.4 ± 40.6	0.249
FA (UI/L)	127.1 ± 6.9	123.7 ± 11.8	117.7 ± 7.5	116.8 ± 13.4	153.1 ± 22.0	0.209
DHL (UI/L)	478.3 ± 26.5	440.2 ± 39.1	446.4 ± 20.8	496.0 ± 52.1	543.4 ± 92.1	0.486
Fibrinogen (mg/dL)	563.9 ± 17.8	674.7 ± 3.8	499.6 ± 25.1 *	600.8 ± 32.8	527.4 ± 59.8	0.001 *
D-dimer (mg/L)	4.6 ± 1.0	6.6 ± 3.4	4.8 ± 1.4	1.6 ± 0.4	5.5 ± 2.8	0.448
PT (seg)	13.2 ± 0.3	13.7 ± 0.4	12.7 ± 0.2	12.8 ± 0.3	14.3 ± 1.2	0.168
INR	1.1 ± 0.0	1.1 ± 0.04	1.0 ± 0.02	1.03 ± 0.0	1.2 ± 0.1	0.109
TTPa (seg)	34.2 ± 1.2	31.5 ± 1.6	31.5 ± 0.9	37.5 ± 4.8	38.6 ± 2.7	0.065
Ferritin (ng/mL)	738.2 ± 74.3	1050.2 ± 290.5	557.0 ± 47.2	767.0 ± 128.3	791.7 ± 176.1	0.098
Procalcitonin (ug/L)	1.4 ± 0.4	0.41 ± 0.05	1.2 ± 0.4	2.5 ± 1.5	1.4 ± 0.7	0.384
IL-6 (pg/mL)	145.7 ± 19.4	155.9 ± 44.3	148.1 ± 28.6	107.4 ± 35.3	184.9 ± 65.1	0.682

Abbreviations: CT, cycle threshold value; BUN, blood urea nitrogen; Na, sodium; K, potassium; CL, chloride; BILT, total bilirubin; BILDir, direct bilirubin; BILInd, Indirect Bilirubin; ProtT, total protein; ALT, alanine aminotransferase; AST, aspartate aminotransferase; FA, alkaline phosphatase; DHL, lactate dehydrogenase; PT, prothrombin time; INR, international normalized ratio; TTPa, activated partial thromboplastin time. * denotes a significant difference at the *p* < 0.05 level.

**Table 3 tropicalmed-08-00134-t003:** Circulating clades of SARS-CoV-2 at hospital admission.

		Frequency (n)	Percentage (%)
Circulating clade	20A	39	19.8
	20B	54	27.4
	20C	13	6.6
	20G	4	2.0
	20I (Alpha, V1)	1	0.5
	20J (Gamma, V3)	7	3.6
	21C (Epsilon)	1	0.5
	21G (Lambda)	7	3.6
	21I (Delta)	7	3.6
	21J (Delta)	11	5.6
	21K (Omicron)	53	26.9
	Total	197	100.0

## Data Availability

The data sets used and/or analyzed in the current study are available from the corresponding author on reasonable request.

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
