# Peer review of "Clinical and Virological Features of SARS-CoV-2 Variants during the Four Waves of the Pandemic in the Mexican Southeast"

_tropicalmed, 2023, doi:10.3390/tropicalmed8030134_

Round 1

Reviewer 1 Report

1) Four waves of Mexico COVID-19 may be concluded in 2-3 waves based on the majority of VOCs in each wave. Please provide reference for the four waves described in Line 87-89.

2) Regarding factors related to symptoms, vaccination status (no vaccine, 1 dose, 2 dose, 2 dose with booster), should be described and analyzed. Because the COVID-19 vaccine availability vary across two years. They are the most important factors contributing to symptoms and strains of virus. 

3) All medical records were reviewed but only 200 PCR positives were selected for SARS-COV-2 whole genome sequencing. The detail analysis of this 200 samples samples should be described according to waves of COVID-19 and other important information for readers to understand the represtentative natures of these 200 samples from all COVID-19 cases.

4) In relation to 3), diagnostic algorithm for COVID-19 in this hospital should be described over the study period. If antigen tests are the primary diagnostic test, please provide detail information when RT-PCR will be used to detect SARS-COV-2. 

Author Response

Dear Reviewer

We thank you for your helpful comments on the manuscript and have edited them to address their concerns.

Reviewer 2 Report

Paper needs a review of English.  Beside this comment,  the paper is acceptable for publication.

Author Response

Dear Reviewer
We thank you for your helpful comments on the manuscript and have made the review of English.

Reviewer 3 Report

This manuscript is interesting focusing on the clinical of SARS-CoV-2 in patients during each wave. However, this manuscript should be adding more parameters to make it more informative and consistent with other publications.

Major concerns.

1. Suggest adding the approval number of the Juan Graham Casasus Hospital's Ethics committee to the text.

2. Do you have a cycle threshold (Ct) value or copy numbers from the RT-PCR?

It is interesting to compare the Ct or Copy number between each wave. The Ancestral or S clade could give a lower Ct or Copy number than other variants.

Moreover, low Ct or high Copy number implied the viral load and seems correlated with severe or life-threatening. You can compare the Ct or Copy number to the Died case in each wave.

3. The most specific clinical manifestations among COVID-19 patients were anosmia and ageusia

Suggest adding both two parameters to Table 1. If it is not available, suggest adding to the limitation of the study.

Minor concerns.

1. If possible, suggest adding the eye redness to Table 1 to make it more information about clinical manifestations.

2. Is this study collects the bicarbonate ion value (Electrolyte)?
Suggest adding the bicarbonate to Table 2.

3. Is this study collect the pO2 or other signs implying hypoxemia?
It is interesting more than showing only dyspnoea.

Comments.

1. Typo line 113, IBM® SPSS® "Statics".

2. Suggest using an abbreviation of the Litre with only one format to make it consistent. This manuscript was found in both "l" and "L" formats. For example, μl and mmol/L.
The "L" is recommended because is a format of the SI unit.

3. Table 3, suggest using only Latin letter without the diacritic. For example, Epsilon and Omicron.

Author Response

(The authors gave the same response as above.)

Round 2

Reviewer 1 Report

I have no further comments. 

Reviewer 3 Report

After reading the revised version, I am satisfied with the improvement.

That is good to describe more parameters that could be more beneficial to understand the clinical characteristics for further wave(s).